# Characterization of Coarse-Grained Heat-Affected Zones in Al and Ti-Deoxidized Offshore Steels

**Henri Tervo [1,\*], Antti Kaijalainen [1], Vahid Javaheri [1], Satish Kolli [1], Tuomas Alatarvas [2], Severi Anttila [3] and Jukka Kömi [1]**

[1] Materials and Mechanical Engineering, Centre for Advanced Steel Research, University of Oulu, P.O. Box 4200, FI-90014 Oulu, Finland; antti.kaijalainen@oulu.fi (A.K.); vahid.javaheri@oulu.fi (V.J.); satish.kolli@oulu.fi (S.K.); Jukka.Komi@oulu.fi (J.K.)

[2] Process Metallurgy Research Unit, Centre for Advanced Steel Research, University of Oulu, P.O. Box 4300, FI-90014 Oulu, Finland; Tuomas.Alatarvas@oulu.fi

[3] SSAB Europe, Rautaruukintie 155, P.O. Box 93, FI-92101 Raahe, Finland; severi.anttila@ssab.com

\* Correspondence: henri.tervo@oulu.fi

**Abstract:** Deterioration of the toughness in heat-affected zones (HAZs) due to the thermal cycles caused by welding is a known problem in offshore steels. Acicular ferrite (AF) in the HAZ is generally considered beneficial regarding the toughness. Three experimental steels were studied in order to find optimal conditions for the AF formation in the coarse-grained heat-affected zone (CGHAZ). One of the steels was Al-deoxidized, while the other two were Ti-deoxidized. The main focus was to distinguish whether the deoxidation practice affected the AF formation in the simulated CGHAZ. First, two different peak temperatures and prolonged annealing were used to study the prior austenite grain coarsening. Then, the effect of welding heat input was studied by applying three cooling times from 800 °C to 500 °C in a Gleeble thermomechanical simulator. The materials were characterized using electron microscopy, energy-dispersive X-ray spectrometry, and electron backscatter diffraction. The Mn depletion along the matrix-particle interface was modelled and measured. It was found that AF formed in the simulated CGHAZ of one of the Ti-deoxidized steels and its fraction increased with increasing cooling time. In this steel, the inclusions consisted mainly of small (1–4 μm) $TiO_x$-MnS, and the tendency for prior austenite grain coarsening was the highest.

**Keywords:** CGHAZ; acicular ferrite; Ti-deoxidized steel; inclusions; microstructure; prior austenite morphology

## 1. Introduction

The opening of new oil fields in ever colder climates increases demands for steels that can withstand the harsh environmental conditions. The steels used offshore are typically manufactured by thermomechanically controlled hot rolling processes (TMCP) that ensure fine-grained microstructure. By utilizing this approach, both high strength and high toughness in the base plate are obtained. Furthermore, the use of moderately low amounts of carbon and other alloying elements enhances weldability. This good combination of strength, toughness and weldability makes the TMCP steels also suitable for other structural use [1].

Offshore steels are typically required to be weldable using submerged arc welding (SAW), gas-metal arc welding (GMAW) or flux-cored arc welding (FCAW) processes, e.g., as specified by the EN 10225-1 [2]. The specifications require that the degradation of material properties such as toughness in the weld heat-affected zone (HAZ) remain tolerable. Especially, the toughness degradation occurring in the coarse-grained heat-affected zone (CGHAZ), the intercritical heat-affected zone (ICHAZ), and the

intercritically reheated coarse-grained heat-affected zone (ICCGHAZ) should be minimized by design. This can be achieved, e.g., by preventing the occurrence of coarse microstructural features such as upper bainite and by controlling the amount of hard and brittle martensitic-austenitic constituents (MA) that may initiate cleavage cracks [3,4].

The occurrence of acicular ferrite (AF) is usually seen as beneficial in the weld metal and in the HAZ, as its fine irregular microstructure provides an interlocking mechanism to crack propagation. Several inclusion types are known to have the ability to nucleate AF [5,6]. The most prominent inclusions are $Ti_2O_3$, $TiO_2$, $(Ti,Mn)_2O_3$, $MnO-Al_2O_3$, and their combinations with MnS and TiN as complex inclusions [5–10]. The latter two tend to nucleate on pre-existing oxides, thereby forming complex mixes of inclusions. As the Mn diffuses to the inclusion, a local depletion of Mn takes place in the immediate vicinity [11–13]. This local depletion of Mn has been reported to increase the ferrite transformation temperature thus encouraging the nucleation around inclusions within grains prior the habitual grain boundary allotriomorphic ferrite [10–17].

The optimal size for the inclusions regarding the AF formation has been reported to vary between 0.25 and 3 μm [5,6,9]. However, as coarse inclusions typically degrade toughness properties, the inclusion size of about 1 μm is generally seen to be more preferable [5]. In addition to size, the fraction of AF correlates with the number density of inclusions smaller than 2 μm [18].

In addition to inclusion characteristics, it is also known that coarse prior austenite grain size (PAGS) promotes the formation of AF [5,6]. Coarser PAGS reduce the total grain boundary area and thus overall mitigates the grain boundary transformation products. In the weld metals, the critical PAGS depending on boron-alloying has been reported to vary between 20 and 60 μm [19]. Boron decreases the critical PAGS required as it inhibits the grain boundary ferrite transformation [19]. It has been suggested that to achieve more than 60% fraction of AF in the CGHAZ, the PAGS should be coarser than 100 μm [20]. In another study, PAGS larger than 250 μm was required to encourage the formation of AF [7]. On the other hand, it has been suggested that the positive effects of increasing PAGS are diminished when the PAGS becomes coarser than 150 μm [5].

Furthermore, cooling rate also affects the formation of AF. The optimal cooling rate is dependent on the chemical composition of the steel. However, in the case of low-carbon steels, a large volume fraction of intragranular ferrite (IGF) that consists of intragranular acicular ferrite (IAF) and intragranular polygonal ferrite (IPF), was detected when the cooling rate was 5 °C/s ($t_{8/5} \approx 60$ s) [13,21]. Another study suggests that the fraction and lath length of AF in CGHAZ increases up to $t_{8/5}$ of 30 s [22]. However, in Ti-deoxidized steels, a wide selection of cooling rates from $t_{8/5} = 5.6$ to 1000 s have been observed to promote AF [22].

A considerable amount of research into AF has been carried out in the past, especially for the weld metal [7–23]. However, the literature dealing with detailed inclusion characteristics and the effects of welding heat input of the phase balance and Mn-depletion in the HAZ are not fully understood. Thus, in this study we attempted to characterize CGHAZ features in three different laboratory steels, one Al-deoxidized and two Ti-deoxidized, which have chemistries that resemble steels used offshore.

## 2. Materials and Methods

### 2.1. Materials and Heat Treatments

The steels used in this study were experimental laboratory melts that resembled TMCP specifications of high strength offshore steels. One of the steels (denoted as Ref) was a conventional Al-deoxidized steel that was used as a reference, while the other two were deoxidized with Ti ($Ti_{high}$ and $Ti_{low}$). The steels were laboratory cast in a vacuum induction furnace as 85 kg ingots. Subsequently, the slabs were soaked at 1200 °C and conventionally hot rolled into 20 mm plates. The compositions of the steels are listed in Table 1. Overall, the chemistries were comparable, except for those of Si, Al, V, and Ti. The differences for Ti-deoxidized steels were the level of Ti and a minor difference in Si. It should be noted that the total oxygen contents remained higher in Ti-deoxidized steels.

**Table 1.** Chemical compositions of the steels investigated (in wt.%, the remainder being Fe) analyzed using an optical emission spectrometer and combustion analysis.

|  | C | Si | Mn | P | S | Al | Nb | V | Ti | N | O | Others |
|---|---|---|---|---|---|---|---|---|---|---|---|---|
| Ref | 0.05 | 0.01 | 1.6 | 0.005 | 0.003 | 0.037 | 0.01 | 0.01 | 0.016 | 0.006 | 0.0023 | Cr, Mo, Cu, |
| $Ti_{high}$ | 0.05 | 0.03 | 1.7 | 0.005 | 0.003 | 0.002 | 0.01 | 0.07 | 0.027 | 0.006 | 0.0080 | Ni in equal |
| $Ti_{low}$ | 0.05 | 0.23 | 1.7 | 0.007 | 0.003 | 0.003 | 0.01 | 0.07 | 0.016 | 0.008 | 0.0047 | proportions |

The HAZ sub-zones were simulated by using a Gleeble 3800 thermomechanical machine. The samples were cylindrical with a length of 36 mm and diameter of 6 mm. The simulated region of this study would most accurately represent CGHAZ in single-pass welding scenarios or the unaltered CGHAZ in multipass welding scenarios.

In the first experimental part, the tendency for PAGS coarsening was examined by holding samples at 1200 °C for 5 min or 1350 °C for 2 min. After the holding time the samples were water quenched. In the second part, different cooling rates similar to practical welding were applied. The Rykalin 3D cooling model was used to derive the temperature-time cooling profiles. Heating rate of 300 °C/s was used, peak temperature was 1350 °C, and the holding time at peak was 0.1 s. Free span distance was 9 mm. The cooling times from 800 to 500 °C ($t_{8/5}$) were calculated to match the heat inputs of 1, 3.5, and 5 kJ/mm and they were 4.8, 16.7, and 23.8 s, respectively. Once the samples cooled below 250 °C, they were allowed to cool in air to room temperature.

*2.2. Mechanical Testing*

Hardness was measured by using a Duramin-A300 (Struers) device under a 100 N load ($HV_{10}$) from the centerline of the surface that contained the transversal direction (TD) and the plate normal (i.e., thickness) direction (ND), and along the simulated CGHAZ sample.

*2.3. Microstructural Characterization*

Initially, the microstructural features of Nital etched specimens were characterized using a Zeiss Ultra Plus field emission scanning electron microscope (FESEM) operated at 5 kV and a Keyence VK-X200 laser scanning confocal microscope (LSCM). However, these images are not presented in the paper. Grain boundary misorientation distribution and average effective grain sizes were measured from the data obtained using an EDAX electron backscatter diffraction (EBSD) system on the FESEM with an acceleration voltage of 15 kV. An area of 88 μm × 88 μm and a step size of 0.2 μm were used for each scan. From the original acquisitions, three pixels were filtered and grains larger than 0.36 μm were used for the grain size calculations. The grain boundaries with misorientation greater than 15° were considered as effective grain boundaries from the point of view of cleavage crack propagation resistance. Equivalent circular diameter (ECD) was used to define the effective grain size. EBSD was used also for the reconstruction of the PAGS otherwise with the same settings, except that the area was extended to 450 μm × 450 μm and the step size was adjusted to 0.75 μm.

2.3.1. Non-Metallic Inclusion Characterization

To characterize the non-metallic inclusions, energy dispersive X-ray spectrometry (EDS) analyses were carried out at 15 kV and 3.5 nA on a JEOL JSM-7000F FESEM. The data were acquired and analyzed using an Oxford INCA software. The working distance was 10 mm and each inclusion was measured for 1 s live time. The inclusions were analyzed from the simulated CGHAZ samples of $t_{8/5}$ = 16.7 s, which should on average describe the studied steels. The analyzed surfaces were cross-sections containing the transverse direction (TD) and normal direction (ND). The acquired inclusion data included information about the number, size, location, shape, and chemical composition of each inclusion. The sizes of the inclusions were determined using their maximum lengths. The minimum inclusion size included in the results was 1 μm.

Inclusions were classified into appropriate classes depicting which phases they contained. Taking into account the EDS analyses of Al, Mn, Ti, N, and S, molar fractions of $Al_2O_3$, MnO, MnS, TiN, and $TiO_2$ components were estimated for each inclusion with a least-square method [24]. Using this method, only the ratios between the considered elements affect the classification, rather than absolute values. Inclusions containing less than 30 wt.% of Al, Mn, Ti, N, and S combined were discarded from the dataset as unclassified. After converting the molar fractions into wt.%, inclusion classes were built based on the combinations of the considered components with a 10 wt.% threshold. Effectively, five components lead to 31 combinations, i.e., inclusion classes.

### 2.3.2. Prior Austenite Reconstruction

As it was practically impossible to reveal the PAGS using common metallography and etching techniques, MATLAB software supplemented with the MTEX toolbox [25] was employed in order to reconstruct the prior austenite grains from the EBSD results. The reconstruction was performed on the basis of the previous works [26–28] through two main steps. Initially, the orientation relationship between the parent austenite and product ferritic phase, i.e., here mainly bainite, was determined using the Kurdjumov–Sachs (K–S) relationship [29] (i.e., $\{111\}_\gamma//\{110\}_\alpha$, $\langle 110 \rangle_\gamma//\langle 111 \rangle_\alpha$). Then, in the second step, the grain map was divided into separate clusters and parent austenite orientation was calculated for each cluster discretely to reconstruct the austenite orientation map and grain structure.

### 2.3.3. IQ Analysis

To accurately characterize and quantify the complex CGHAZ microstructures, EBSD image quality (IQ) analysis was used. As different morphologies vary in intrinsic dislocation density and give different IQ values [30], the normalized IQ data were analyzed on the basis of previous work by DeArdo [31]. In this method, the normalized IQ histogram was deconvoluted into multiple peaks with a normal distribution and then the ratio of area under each peak to the area of IQ curve was considered as a fraction associated with each morphology or phase.

### 2.3.4. Mn Depletion around Inclusions

For the observation of a single inclusion and a local Mn depletion in the steel matrix next to the inclusion, a 200 kV energy filtered scanning transmission electron microscope (JEOL JEM-2200FS EFTEM/STEM) (JEOL Ltd., Akishima, Kanto, Japan) was used in the transmission and scanning transmission (STEM) modes. Thin foils for TEM studies were prepared using the FEI Helios Dual Beam Focused Ion Beam (FIB) system with an accelerating voltage of 30 kV and beam currents in the range 90 pA to 9 nA. An ion beam was also used for thinning and polishing the FIB lamella.

### 2.4. Modelling

The MnS precipitates were nucleated on the inclusions and could be expected to grow with the aid of diffusion from the matrix into the precipitates. For the calculations, it was assumed that the MnS precipitates were already nucleated and the growth of these precipitates was modelled using the diffusion module in Thermo-Calc (Thermo-Calc Software AB, Stockholm, Sweden), i.e., DICTRA. This is a tool for simulating the diffusion kinetics in multicomponent alloys [32]. It was also assumed that the MnS is formed continuously over the oxides, thereby reducing the problem to one dimension. The size of the cell for measurement was considered arbitrarily to be one-tenth of the grain size. The local equilibrium was assumed at the austenite matrix–MnS interface. The simulations were based on a numerical solution of multicomponent diffusion equations that were treated by a combination of MnS composition predictions and local equilibrium conditions at the interface. TCFE7 thermodynamic database and MOBFE2 mobility databases were used for performing the simulations.

## 3. Results and Discussion

### 3.1. Inclusions

#### 3.1.1. Size Distribution

The size distributions of all inclusions in the $t_{8/5}$ = 16.7 s samples are presented in Figure 1. The majority of inclusions in the Ref sample were small 1–2 µm, while in $Ti_{high}$ there were higher fraction of coarser inclusions (length > 2 µm). This is shown especially well in Figure 1b, wherein the inclusions are presented as area fractions. The larger area fraction of coarser particles is explained by the differences seen in the total oxygen content between the samples. The Al-deoxidized Ref had an oxygen content of 23 ppm, whereas the Ti-deoxidized steels had 47 and 80 ppm. The inclusion size distribution in Ref was close to that of $Ti_{low}$ but the average size was slightly larger in $Ti_{low}$.

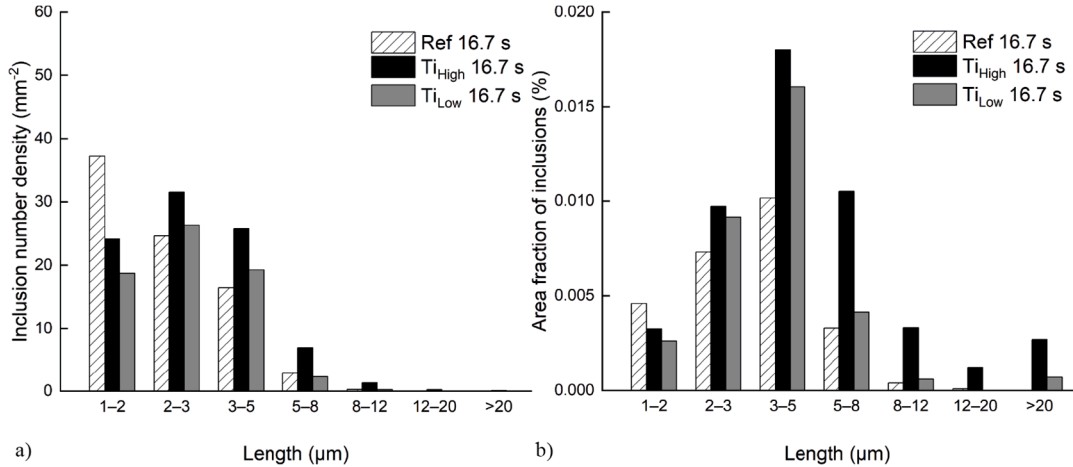

**Figure 1.** Size distributions of inclusions in the $t_{8/5}$ = 16.7 s samples presented as number densities (**a**) and as area fractions (**b**).

#### 3.1.2. Inclusion Types

In practice, inclusions can contain various titanium oxide phases, such as $TiO_2$ or $Ti_2O_3$, but here $TiO_x$ was used to denote all of them. The number of inclusion classes was reduced by combining relevant types. Table 2 shows the inclusion classification obtained by the used method and the number density of inclusions in the studied samples accordingly. As seen in the table, there were a lot of MnO-$TiO_x$(+TiN) and MnO-$TiO_x$-MnS(+TiN) inclusions in the $Ti_{high}$ and $Ti_{low}$ samples. Pure TiN inclusions were spotted in the Ref sample, while in Ti-deoxidized samples their number density was close to zero. However, TiN was combined with other inclusion types in $Ti_{high}$ and $Ti_{low}$.

Figure 2 presents the number densities of different class of inclusions in the $t_{8/5}$ = 16.7 s samples according to the classification scheme used. It can be clearly seen that the majority of the inclusions in $Ti_{high}$ and $Ti_{low}$ were $TiO_x$-containing, while inclusions containing considerable amounts of TiN (i.e., TiN or MnS TiN classes) were common only in the Ref and $Ti_{high}$ samples. A high number of $Al_2O_3$-containing inclusions in the Ref sample indicates that the most common inclusions were different than in the other two steels, as was also pointed out in Table 2.

**Table 2.** Number density of small (<3 µm) inclusions per mm$^2$ in the studied samples.

| Inclusion Type | Combined Class | Ref 16.7 s (<3 µm) | Ref 16.7 s (all) | Ti$_{High}$ 16.7 s (<3 µm) | Ti$_{High}$ 16.7 s (all) | Ti$_{Low}$ 16.7 s (<3 µm) | Ti$_{Low}$ 16.7 s (all) |
|---|---|---|---|---|---|---|---|
| Al$_2$O$_3$ | Al$_2$O$_3$ containing oxides | 2.2 | 3.6 | – | – | – | – |
| Al$_2$O$_3$ MnO | Al$_2$O$_3$ containing oxides | 0.2 | 0.2 | – | – | – | – |
| Al$_2$O$_3$ MnO MnS | Al$_2$O$_3$ containing complex | 0.1 | 0.1 | – | – | – | – |
| Al$_2$O$_3$ MnO MnS TiN | Al$_2$O$_3$ containing complex | 1.8 | 1.8 | – | – | 0.3 | 0.3 |
| Al$_2$O$_3$ MnO TiN | Al$_2$O$_3$ containing complex | 4.4 | 4.6 | – | – | – | – |
| Al$_2$O$_3$ MnO TiO$_x$ | Al$_2$O$_3$ containing oxides | – | – | – | – | 0.7 | 1.1 |
| Al$_2$O$_3$ MnO TiO$_x$ MnS | Al$_2$O$_3$ containing complex | 0.1 | 0.1 | – | – | 0.2 | 0.2 |
| Al$_2$O$_3$ MnO TiO$_x$ MnS TiN | Al$_2$O$_3$ containing complex | 0.1 | 0.1 | – | – | 0.4 | 0.5 |
| Al$_2$O$_3$ MnO TiO$_x$ TiN | Al$_2$O$_3$ containing complex | 1.3 | 1.3 | – | – | 0.6 | 1.0 |
| Al$_2$O$_3$ MnS | Al$_2$O$_3$ containing complex | 1.2 | 2.5 | – | – | – | – |
| Al$_2$O$_3$ MnS TiN | Al$_2$O$_3$ containing complex | 4.5 | 6.3 | – | – | – | – |
| Al$_2$O$_3$ TiN | Al$_2$O$_3$ containing complex | 9.5 | 13.5 | – | – | – | – |
| Al$_2$O$_3$ TiO$_x$ | Al$_2$O$_3$ containing oxides | 0.5 | 1.0 | – | – | 0.1 | 0.2 |
| Al$_2$O$_3$ TiO$_x$ MnS | Al$_2$O$_3$ containing complex | 0.4 | 0.6 | – | – | – | – |
| Al$_2$O$_3$ TiO$_x$ MnS TiN | Al$_2$O$_3$ containing complex | 0.3 | 0.4 | – | – | – | – |
| Al$_2$O$_3$ TiO$_x$ TiN | Al$_2$O$_3$ containing complex | 3.0 | 3.7 | – | – | – | – |
| MnO | MnO(+MnS-TiN) | – | – | – | – | – | – |
| MnO MnS | MnO(+MnS-TiN) | 0.1 | 0.1 | 0.1 | 0.2 | 1.2 | 1.2 |
| MnO MnS TiN | MnO(+MnS-TiN) | 3.6 | 3.7 | 2.8 | 3.1 | 6.9 | 7.2 |
| MnO TiN | MnO(+MnS-TiN) | 16.5 | 17.5 | 5.9 | 7.7 | 1.8 | 2.1 |
| MnO TiO$_x$ | MnO-TiO$_x$(+TiN) | – | – | 14.6 | 33.5 | 5.5 | 14.9 |
| MnO TiO$_x$ MnS | MnO-TiO$_x$-MnS(+TiN) | – | – | 5.2 | 9.7 | 4.7 | 6.3 |
| MnO TiO$_x$ MnS TiN | MnO-TiO$_x$-MnS(+TiN) | – | – | 5.9 | 7.1 | 9.3 | 11.2 |
| MnO TiO$_x$ TiN | MnO-TiO$_x$(+TiN) | 1.0 | 1.0 | 10.3 | 11.4 | 7.0 | 9.2 |
| MnS | MnS | 2.8 | 4.2 | 1.9 | 3.7 | 0.5 | 0.7 |
| MnS TiN | MnS TiN | 2.9 | 4.8 | 4.4 | 6.1 | 1.7 | 2.5 |
| TiN | TiN | 4.7 | 9.3 | 0.2 | 0.5 | 0.5 | 1.4 |
| TiO$_x$ | TiO$_x$(+MnS-TiN) | – | – | 0.3 | 1.0 | 0.9 | 3.4 |
| TiO$_x$ MnS | TiO$_x$(+MnS-TiN) | 0.3 | 0.3 | 1.7 | 3.3 | 1.1 | 1.3 |
| TiO$_x$ MnS TiN | TiO$_x$(+MnS-TiN) | 0.1 | 0.1 | 1.9 | 2.4 | 1.0 | 1.3 |
| TiO$_x$ TiN | TiO$_x$(+MnS-TiN) | 0.4 | 0.5 | 0.5 | 0.5 | 0.5 | 0.7 |

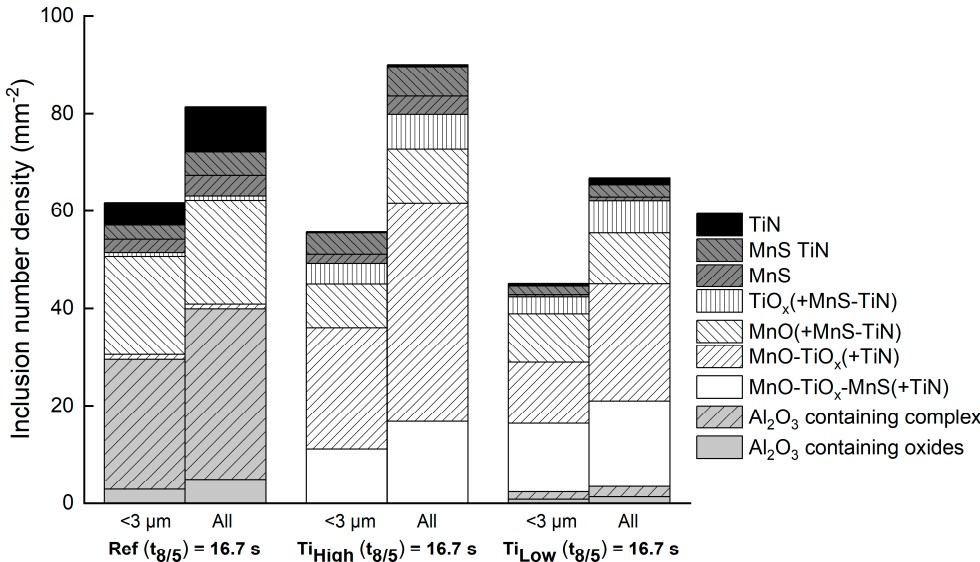

**Figure 2.** Number density and classification of inclusions smaller than 3 μm and all inclusions in the studied steels.

Inclusions smaller than 3 μm are presented separately, as these small inclusions were considered to be in the key role for inducing AF formation, while the coarser inclusions may contribute little to AF and cause degradation of toughness and ductility. In this study, the most promising inclusions regarding the AF formation should be those consisting of MnO-$TiO_x$ together with MnS regardless of the occurrence of TiN, i.e., class MnO-$TiO_x$-MnS(+TiN) in Figure 2. This is based on the assumption that the nucleation energy for the heterogenic ferrite nucleation is smaller at the interface of $TiO_x$ or MnO-$TiO_x$ and austenitic steel matrix than at the austenite grain boundary. In addition to this, local Mn content in the austenite around inclusions was expected to decrease when Mn either diffused to $TiO_x$ forming MnO-$TiO_x$ or precipitated as MnS on the surface of the oxide. Additionally, this Mn depletion around inclusion induced ferrite nucleation on the inclusion instead of grain boundary due to local increase in the start temperature of ferrite formation. Thus, these inclusions have a doubly beneficial effect on the AF formation. As seen in Figure 2, the inclusions of this type made up roughly 1/3 of all inclusions smaller than 3 μm in $Ti_{low}$, while in $Ti_{high}$ their fraction was clearly smaller. However, in $Ti_{high}$ the majority of inclusions belonged to MnO-$TiO_x$(+TiN) class, which may also have similar effects on AF formation.

### 3.2. Growth of Prior Austenite Grain Size

As an increase of PAGS is known to encourage the AF formation, one part of the study was to intentionally increase the PAGS. For reference, PAGS was measured also from the base materials. Reconstructed prior austenite grains with the mean PAGS in the studied steels in base material and after heating to the peak temperatures of 1200 °C (for 5 min) and 1350 °C (for 2 min) and subsequently water quenched are presented in Figure 3. The PAGS distribution of each variant is presented separately in Figure 4.

Figures 3 and 4 show that the PAGSs for the sample with $T_p$ = 1200 °C were still relatively small and followed a normal distribution in all samples, indicating moderate grain growth. The smallest PAGS was detected in the Ref, while the largest PAGS was in $Ti_{low}$. Conversely, for the samples held with the $T_p$ = 1350 °C, the PAGSs were noticeably larger in all steels such that microstructure consisted of many grains greater than 100 μm in both Ref and $Ti_{high}$. Curiously, in one $Ti_{low}$ the EBSD sample area of the PAGS increased enormously and only a few whole grains were captured even when an area was built from six separate EBSD runs (resulted an image size of 0.4 × 2.4 mm). The enormous prior austenite grain growth in $Ti_{low}$ may suggest the lack of nanoscale precipitates such as TiN, which are

known to inhibit the grain growth through the pinning effect. However, TiN was detected at least in microscale inclusions also in Ti$_{low}$, e.g., together with MnO-TiO$_x$ and MnO-TiO$_x$-MnS inclusions, as was presented in Table 2. It is still evident that in Ref and Ti$_{high}$ steels the fractions of inclusions containing considerable amounts of TiN, such as pure TiN and MnS-TiN and MnO-TiN, were higher than in the Ti$_{low}$ steel. This may mean that the amounts of nanoscale TiN were also higher in Ref and Ti$_{high}$ than in Ti$_{low}$ and that would explain the large PAGS in Ti$_{low}$ compared to other samples.

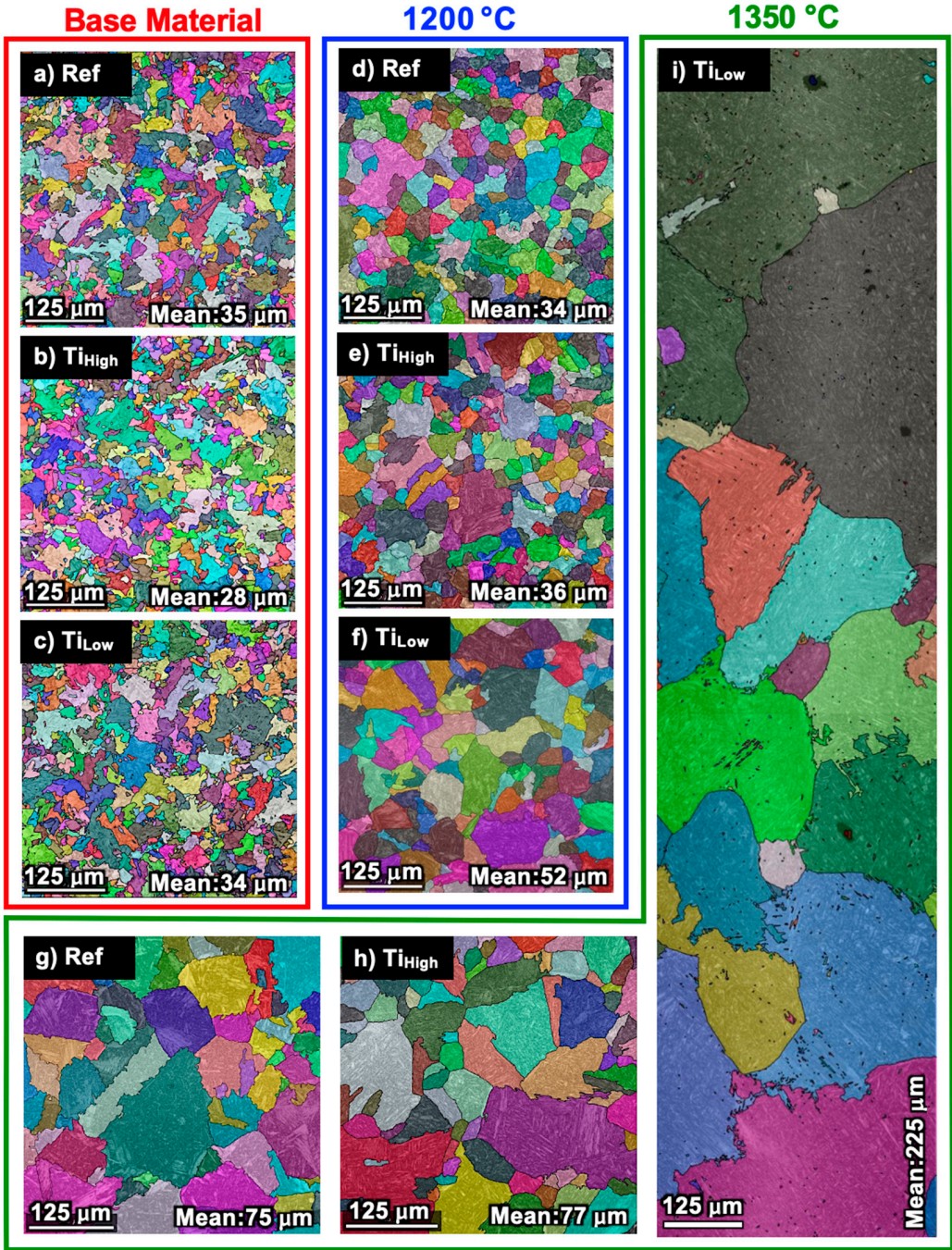

**Figure 3.** Reconstructed prior austenite grains with the mean prior austenite grain size of base materials of Ref (**a**), Ti$_{high}$ (**b**), and Ti$_{low}$ (**c**); with T$_P$ = 1200 °C, held for 5 min, and water quenched Ref (**d**), Ti$_{high}$ (**e**), and Ti$_{low}$ (**f**); and with T$_P$ = 1350 °C, held for 2 min, and water quenched Ref (**g**), Ti$_{high}$ (**h**), and Ti$_{low}$ (**i**). Images are from TD-ND surface of the samples, so possible elongation regarding the rolling direction is not visible. Note that in (**i**) an area matrix of 1 × 6 was combined to one image to capture some grain boundaries.

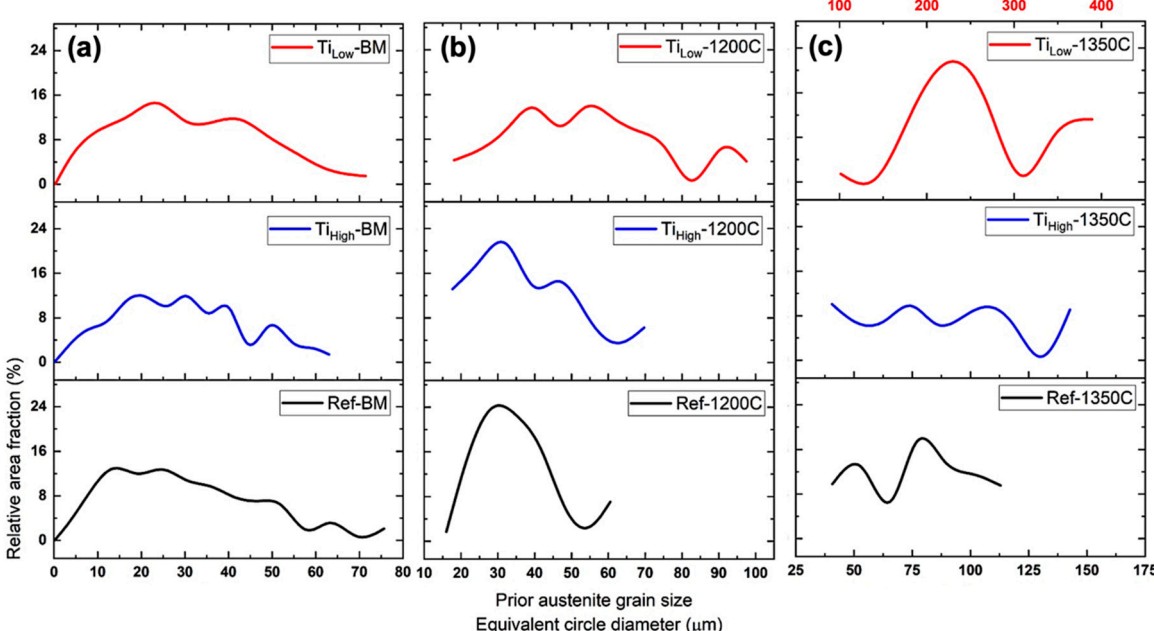

**Figure 4.** (**a**) Prior austenite grain size distribution of base materials; (**b**)simulated HAZ with $T_p$ = 1200 °C, held for 5 min and water quenched; (**c**) and simulated HAZ with $T_p$ = 1350 °C, held for 2 min, and water quenched. Note the differences in the *x*-axis. The *x*-axis for the $Ti_{low}$ −1350 °C is given in red on the top of the chart.

### 3.3. Microstructure of Simulated CGHAZ with Different Cooling Rates

To carry out a detailed study of the microstructure and to compare the fractions of each microstructural component in the studied CGHAZ samples, EBSD-IQ data were analyzed, and the results are presented in Figures 5–7. For the calculations, the locations of different phases and morphologies with the normalized IQ values (from 0 to 100) were considered as follows: (polygonal) ferrite 90, plate-like bainite 70, acicular ferrite 60, granular bainite 50, lath-like bainite 35, and martensite 25.

The majority of all CGHAZ microstructures consisted of plate-like bainite (44–68%). However, it can be observed that granular bainite, plate-like bainite, and lath-like bainite morphologies were the dominate microstructural constituents in the Ref and $Ti_{high}$, whereas granular bainite was replaced by AF (19–28%) in the $Ti_{low}$.

The cooling rates used caused only slight changes in the microstructures. Most noticeably in the $t_{8/5}$ = 23.8 s scenario (Figure 7) the fraction of AF in $Ti_{low}$ increased to 28%, while the majority (51%) was still determined as plate-like bainite. The reason for the slight increase in the fraction of AF with increasing cooling time in $Ti_{low}$ stems from the continuous cooling transformation (CCT) diagram; the transformation of AF occurred between bainite and polygonal ferrite [33]. Thus, in the current study the increase in the cooling time reduced the driving force for the bainite transformation promoting AF transformation. An example of AF growing from an inclusion is visible in Figure 7e. In the corresponding Ref and $Ti_{high}$ the main microstructural components were still plate-like bainite and granular bainite.

The validity of the IQ analysis for separating the AF from the other microstructural components was checked with IQ data from an autogenous weld metal sample that had AF as a dominate constituent. The results presented in Figure 8 verified the assumption. It should be remarked that high IQ value phase indicates several features present in the microstructure, such as the martensite-austenite (MA) islands, inclusions, pitting region caused by sample preparation, and lath-like morphology.

As the grain sizes had non-normal distributions, median and 80% cumulative grain sizes were used and are listed in Table 3. Generally, the median varied from 1.22 to 2.44 μm. Unsurprisingly,

the longer cooling time coarsened the grain size. With $t_{8/5}$ = 4.8 s both the coarsest grains and the median were finer in $Ti_{low}$ compared to other steels ($p < 0.05$, Mann–Whitney). In the intermediate $t_{8/5}$ = 16.7 s the statistical differences were only seen between the medians of steels $Ti_{low}$ and $Ti_{high}$. Finally, with $t_{8/5}$ = 23.8 there were statistical differences, except between Ref and $Ti_{low}$.

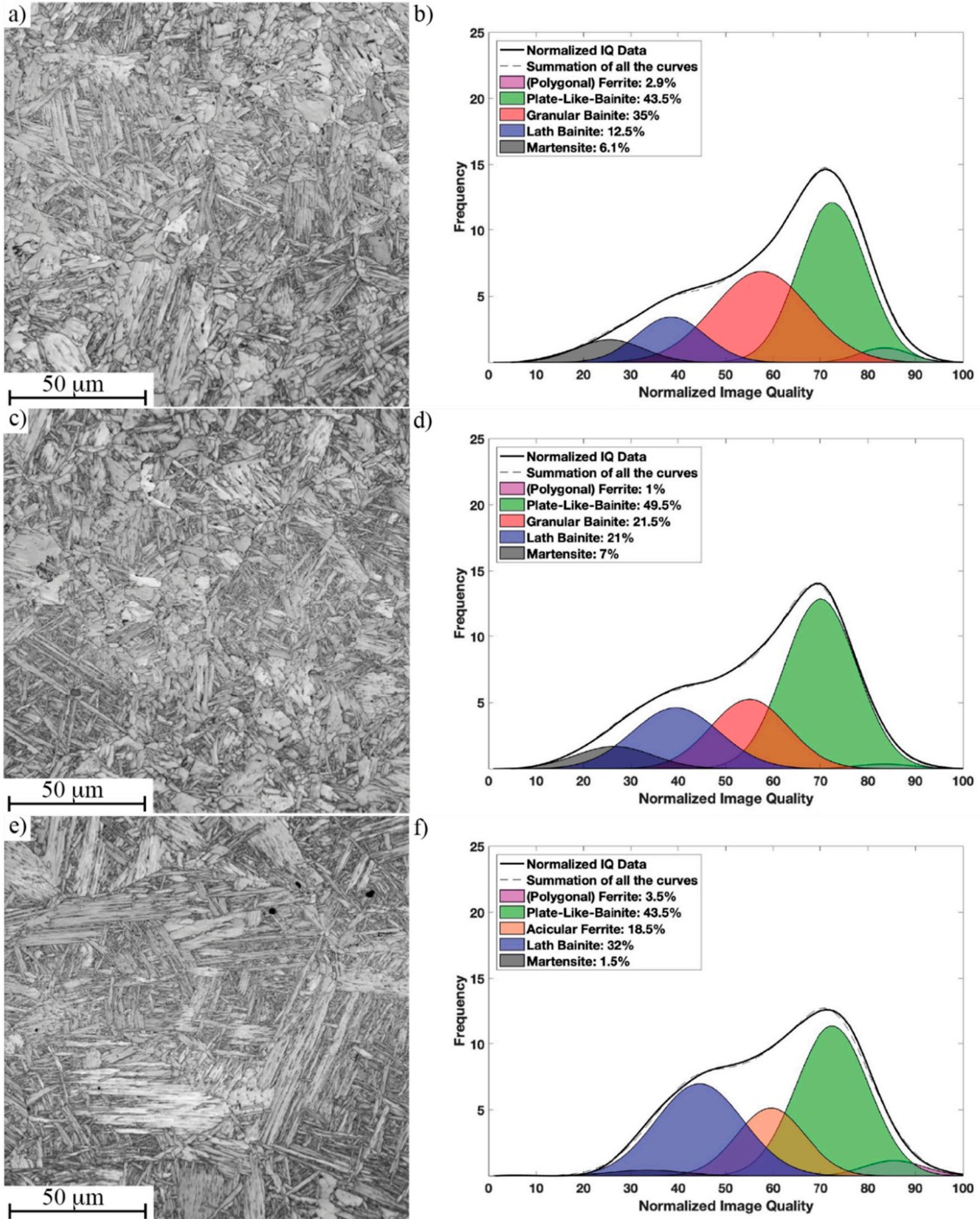

**Figure 5.** IQ analysis of simulated CGHAZ with $t_{8/5}$ = 4.8 s in Ref (**a**,**b**), $Ti_{high}$ (**c**,**d**), and $Ti_{low}$ (**e**,**f**).

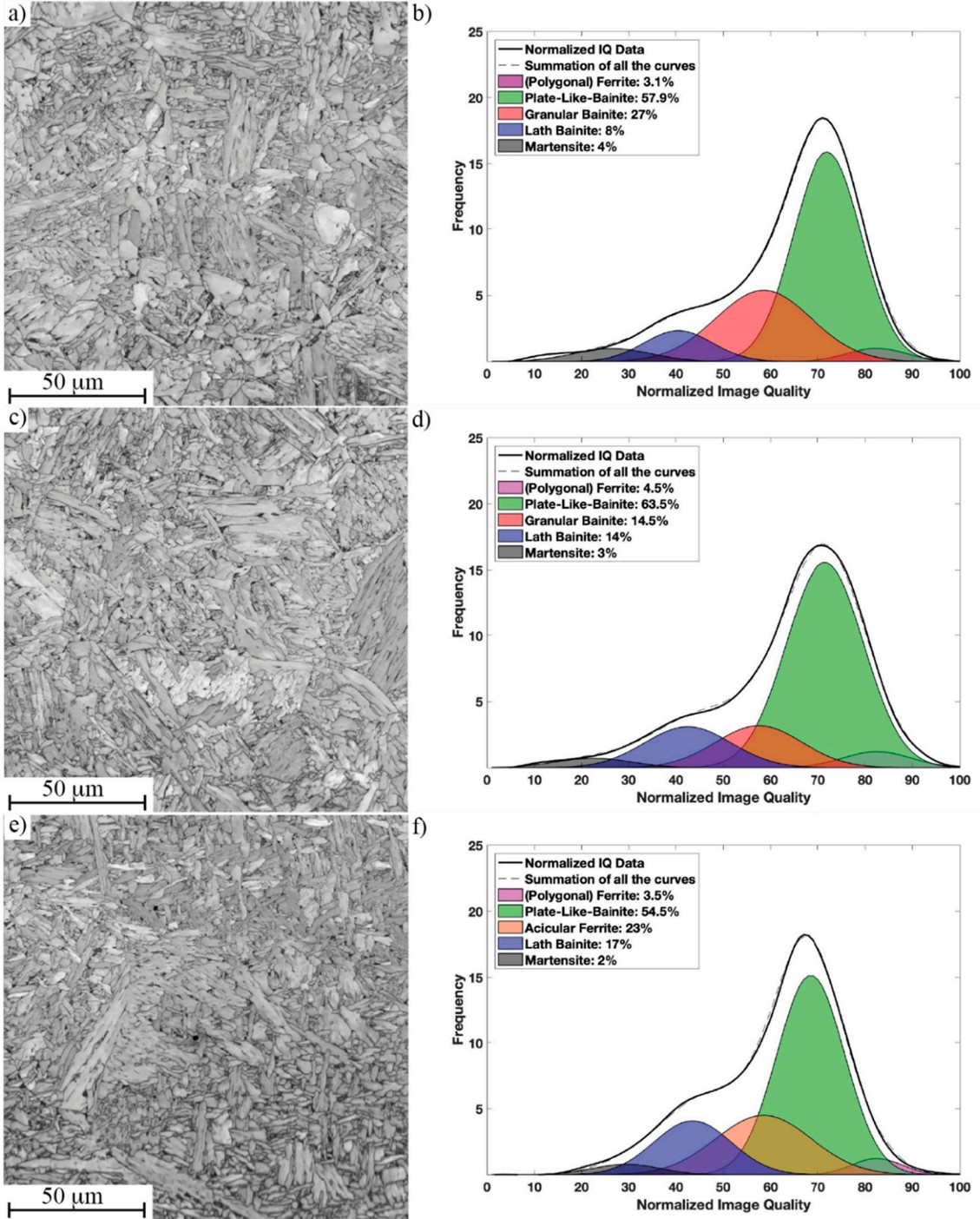

**Figure 6.** IQ analysis of simulated CGHAZ with $t_{8/5}$ = 16.7 s in Ref (**a**,**b**), Ti$_{high}$ (**c**,**d**), and Ti$_{low}$ (**e**,**f**).

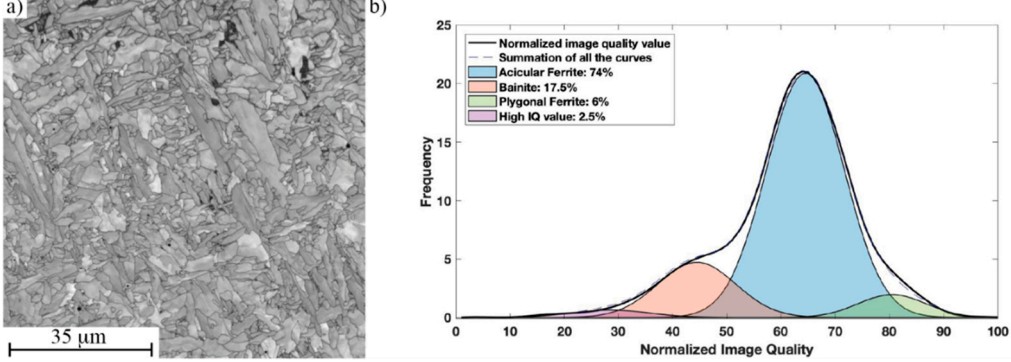

**Figure 7.** IQ analysis of simulated CGHAZ with $t_{8/5}$ = 23.8 s in Ref (**a**,**b**), $Ti_{high}$ (**c**,**d**), and $Ti_{low}$ (**e**,**f**).

**Figure 8.** IQ analysis of weld metal in $Ti_{low}$ with $t_{8/5}$ = 23.8 s. (**a**) EBSD-image and (**b**) normalized IQ values.

**Table 3.** Median grain size and grain size at 80% in the cumulative grain size distribution of the studied steels with $t_{8/5}$ = 4.8 s, 16.7 s, and 23.8 s.

| Grain Size | Ref 4.8 s | Ti$_{high}$ 4.8 s | Ti$_{low}$ 4.8 s | Ref 16.7 s | Ti$_{high}$ 16.7 s | Ti$_{low}$ 16.7 s | Ref 23.8 s | Ti$_{high}$ 23.8 s | Ti$_{low}$ 23.8 s |
|---|---|---|---|---|---|---|---|---|---|
| Median (µm) | 1.54 | 1.37 | 1.22 | 1.92 | 1.73 | 2.06 | 1.84 | 2.44 | 1.61 |
| D80% (µm) | 14.27 | 11.20 | 8.92 | 16.43 | 19.27 | 14.75 | 21.71 | 20.71 | 17.49 |

Misorientation distribution of Ti$_{low}$ differed from the other steels, as seen in Figure 9. The misorientation distribution of Ti$_{low}$ resembled that of AF presented in a previous study [34], especially in cases with $t_{8/5}$ = 16.7 s and 23.8 s. In Ref and Ti$_{high}$ the misorientation distributions were comparable with each other. Figure 9 presents the microstructures and the HV10 hardness of each scenario. With $t_{8/5}$ = 4.8 s the average HV10 hardness in Ti$_{low}$ was notably higher (280) than in the other two. Potentially, this might be a result of an increased hardenability due to coarse PAGS, as was seen, e.g., in Figure 3i and increased lath-like bainitic features seen in IQ analysis. However, as the prior IQ analysis shows, the fraction of martensite was not seen to increase. Another factor improving hardenability in Ti$_{low}$ compared to other steels was higher silicon content (0.23% vs. 0.01% and 0.03% in Ref and Ti$_{high}$, respectively). With longer cooling times the differences in hardness diminished, but the trend remained, i.e., the Ref was the softest and Ti$_{low}$ the hardest ($p < 0.05$, one-way ANOVA).

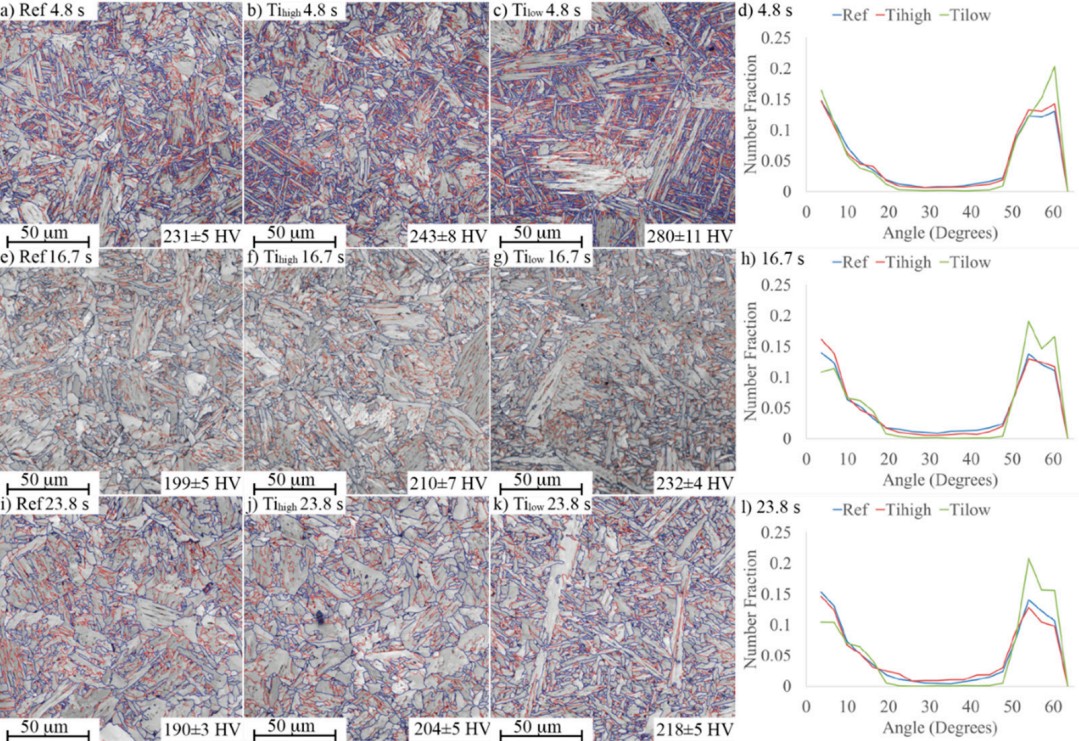

**Figure 9.** Grain boundary misorientation distributions of the simulated CGHAZ of the studied steels using $t_{8/5}$ = 4.8 s (**d**), 16.7 s (**h**), and 23.8 s (**l**) together with the EBSD-images and average hardness values of the corresponding variants. From left to right Ref, Ti$_{high}$ and Ti$_{low}$; from top to bottom $t_{8/5}$ = 4.8 s, 16.7 s, and 23.8 s, respectively. Red lines in the microstructure images indicate the low-angle boundaries 2–15°, while the blue lines indicate high-angle boundaries greater than 15°. (**a**) Ref 4.8 s, (**b**) Ti$_{high}$ 4.8 s, (**c**) Ti$_{low}$ 4.8 s, (**e**) Ref 16.7 s, (**f**) Ti$_{high}$ 16.7 s, (**g**) Ti$_{low}$ 16.7 s, (**i**) Ref 23.8 s, (**j**) Ti$_{low}$ 23.8 s and (**k**) Ti$_{high}$ 23.8 s.

### 3.4. TEM Studies of Mn Depletion around Inclusions

A single (3–5 μm) $TiO_x$-MnO+TiN inclusion and the steel matrix around it in the simulated CGHAZ of $Ti_{low}$ with $t_{8/5}$ = 23.8 s were studied using TEM, and the main results are presented in Figure 10. Figure 10a shows approximately half of the cross section of the inclusion as the upper edge of the inclusion is the polished surface in the original steel sample. Compared to SEM observations, the TEM observations revealed more detailed features regarding different phases present in and around one inclusion. Smaller phases, e.g., MnS in this case, had so small fraction of the whole inclusion such that these would be hardly observable in ordinary SEM inclusion characterization. The captured inclusion represented the optimal type of inclusion regarding the AF formation based on chemistry, although the size of the particle as a whole was slightly coarser than optimal.

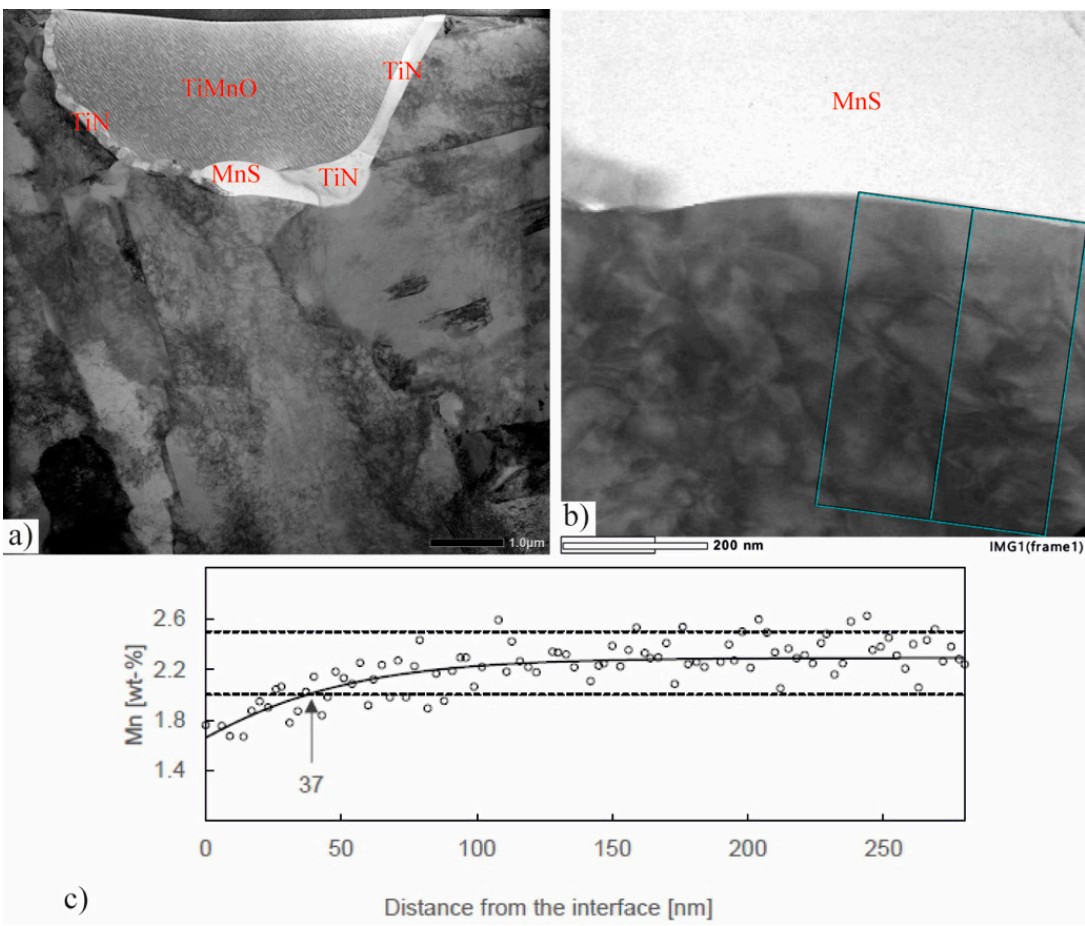

**Figure 10.** (**a**) TEM image of a typical inclusion in the CGHAZ of $Ti_{low}$ with $t_{8/5}$ = 23.8 s; (**b**) zoomed area where the Mn-measurement line is indicated; (**c**) the measured Mn content with dotted lines that represent 2 standard deviations from the mean taken from data points >200 nm from the interface.

The main aim of the TEM study was to measure the Mn content at the interface between the steel matrix and MnS. Figure 10b presents the line whereat the Mn was measured using EDS, and Figure 10c shows how the Mn content gradually decreased from approximately 2.2 wt.% to 1.7 wt.% during the distance of about 50–60 nm. Taking into account the scattering of the data, the depletion depth in this instance was about 37 nm from the interface (the point in which the fitted trendline deviated outside the two standard deviations of the bulk). It was noticed that the bulk Mn content was different in EDS as compared to bulk material OES in $Ti_{low}$. The difference was assumed to come from local segregation of Mn and differences in analysis techniques.

*3.5. DICTRA Simulations*

The concentration profiles of Mn across the MnS–austenite matrix interface for the considered cooling times to simulate CGHAZ in this study was calculated. These depletion profiles are shown schematically in the Figure 11, from the interface of MnS–austenite and into the austenite matrix. The concentrations at zero nm indicate the Mn composition at the interface in austenite matrix. The profiles in Figure 11a were obtained after a cooling from 800 to 500 °C ($t_{8/5}$) of 4.8, 16.7, and 23.8 s. A clear depletion of Mn was observed, and the depletion width increased with an increase in $t_{8/5}$. This in turn may explain, partly, the increase of the AF fraction in the CGHAZ of $Ti_{low}$ with increasing cooling time, as was presented in Figures 5–7; the driving force for the ferrite formation is known to increase along with decreasing Mn content. Similar observations have been achieved previously, e.g., in [11–13,17]. The depletion of Mn is in the nm range for the considered conditions. However, the alloy might consist of Mn-depleted regions prior the start of simulations, and because of this, calculations were also performed assuming an initial drop of Mn to 1.6 by wt.% at the interface. These are shown in Figure 11b. The depletion of Mn increased in width, although still remaining in the nm range. These results suggest the presence of Mn depletion and estimating approximate widths and are in conjunction with that seen from the TEM measurements.

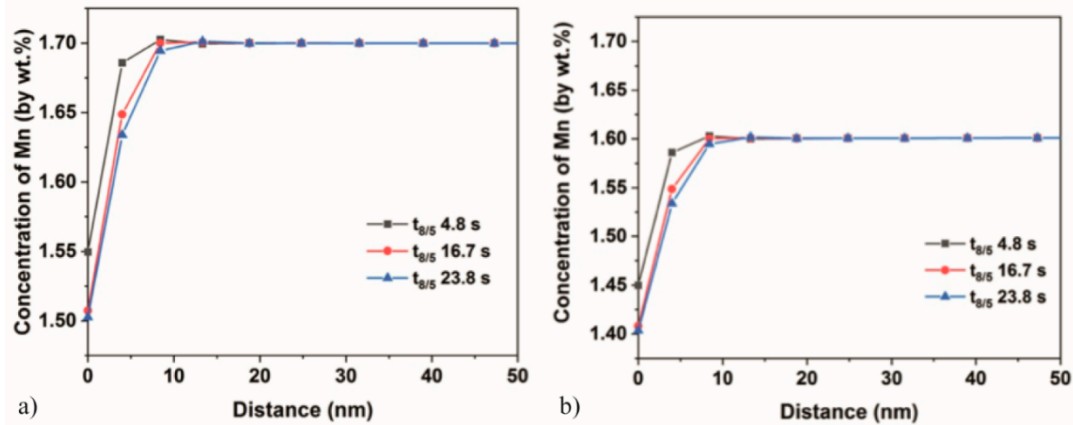

**Figure 11.** Mn concentration profiles from MnS–austenite matrix interface into the austenite matrix for the cooling times used for CGHAZ simulation in this study (**a**) assuming no initial Mn depletion and (**b**) assuming an initial depletion of Mn to 1.6 wt.% at the interface.

## 4. Conclusions

In this study, various factors affecting the AF formation in experimental laboratory steels' CGHAZ was investigated. The following conclusions can be drawn from the results.

The most optimal inclusion content for AF was achieved in this study with the $Ti_{low}$ steel, while in the $Ti_{high}$ steel the inclusions were mostly too coarse, likely owing to high O and Ti contents. The latter also promoted TiN containing inclusions. This observation emphasizes the importance of Ti control; there should be enough Ti to form $TiO_x$ but not too much to prevent the formation of TiN. In Ref, the inclusions were typical for those of Al-deoxidized steels, mainly $Al_2O_3$, Mns, TiN, and combinations of them. However, it should be noted that in industrial-scale Ca-treatment should also have been included, which would change the inclusions by replacing $Al_2O_3$ and MnS inclusions with $CaO–Al_2O_3$, CaS and $CaO–Al_2O_3$–CaS inclusions.

The DICTRA simulation suggested and the TEM study confirmed the depletion of manganese in the steel matrix close to $MnO-TiO_x(+MnS)$ inclusions. Local decrease of Mn in the austenite is known to increase locally the start temperature for the ferrite and thus promote the intragranular AF formation. As the majority of the inclusions detected in $Ti_{low}$ consisted of $MnO-TiO_x(+MnS)$ compounds, it is likely that the presence of these inclusions and the AF detected in the CGHAZ have a relation.

PAGS was observed to have grown, especially in $Ti_{low}$, typically for a CGHAZ simulation; some individual PAGS were measured in several hundreds of micrometers. The growth of PAGS in $Ti_{low}$ may have been a result of the lack of TiN precipitates that would otherwise inhibit the austenite grain growth. Microscale TiN inclusions occurred also in this steel but mainly combined with other types of inclusions. Thus, the reason for the grain growth only in $Ti_{low}$ requires more studies. Again, the AF was only detected in CGHAZ of $Ti_{low}$, which had a significantly greater PAGS coarsening tendency compared to other steels. It was observed that the fraction of AF increased with increasing cooling time, but only somewhat modest amounts (19–28%) were seen. Thus, it might be that a longer cooling time would still be required to form more substantial amount of AF. The absence of AF in $Ti_{high}$ requires more studies, but some explanation for the difference could be different inclusion content and/or smaller PAGS than in $Ti_{low}$.

Main microstructural constituents together with corresponding hardness values of each studied material variant are summarized in Table 4.

**Table 4.** Summary of the key material performance measures. PB = plate-like bainite, AF = acicular ferrite, GB = granular bainite, LB = lath bainite.

| Performance | $t_{8/5}$ = 4.8 s Microstructure/Hardness (HV10) | $t_{8/5}$ = 16.7 s Microstructure/Hardness (HV10) | $t_{8/5}$ = 23.8 s Microstructure/Hardness (HV10) |
|---|---|---|---|
| Ref | PB-GB/231 ± 5 | PB-GB/199 ± 5 | PB-GB/190 ± 3 |
| $Ti_{high}$ | PB-GB/243 ± 8 | PB-GB/210 ± 7 | PB-GB/204 ± 5 |
| $Ti_{low}$ | PB-LB/280 ± 11 | PB-AF/232 ± 4 | PB-AF/218 ± 5 |

**Author Contributions:** Conceptualization, H.T., A.K., and S.A.; methodology, T.A., V.J., and S.K.; formal analysis, H.T., S.A., T.A., and V.J.; investigation, H.T. and S.A.; resources, S.A., T.A., V.J., and S.K.; data curation, H.T.; writing—original draft preparation, H.T., S.A., V.J., S.K., and T.A.; writing—review and editing, H.T., A.K., S.A., V.J., and T.A.; visualization, H.T., S.A., V.J., S.K., and T.A.; supervision, A.K. and J.K.; project administration, A.K. and J.K.; funding acquisition, J.K. All authors have read and agreed to the published version of the manuscript.

**Funding:** The authors are grateful to Business Finland for financing this work as a part of the research project ISA–Intelligent Steel Applications.

**Acknowledgments:** SSAB Europe is acknowledged for the studied materials. Ilpo Alasaarela, Juha Uusitalo and Tun Tun Nyo are acknowledged for sample preparation and Gleeble simulation.

**Conflicts of Interest:** The authors declare no conflict of interest.

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
