# Peer review of "Characterization of Coarse-Grained Heat-Affected Zones in Al and Ti-Deoxidized Offshore Steels"

_metals, doi:10.3390/met10081096_

Round 1

Reviewer 1 Report

This manuscript is not suitable as a research paper because this only reports experimantal results without deep discussion. For example, authors should discuss the reason why the fraction of AF increased with increasing cooling time in Tilow sample. Please consider the change of the driving force when the local decrease of Mn was observed.

Author Response

Dear Editor,

The authors would like to thank the editor and the reviewers for their valuable comments and suggestions to improve the above manuscript. The response of the authors to the individual comments is given below and we hope that the changes are sufficient for the manuscript to be given final approval.

The amendments in the manuscript are marked using Track Changes function.

Reviewer’s comments:

Reviewer #1

Point 1: This manuscript is not suitable as a research paper because this only reports experimantal results without deep discussion. For example, authors should discuss the reason why the fraction of AF increased with increasing cooling time in Tilow sample. Please consider the change of the driving force when the local decrease of Mn was observed.

Response 1: We don't have separate "Discussion" chapter in the paper but considerable discussion and speculation is provided together with the results under the title "Results and discussion". The role of Mn depletion zone was discussed in the inclusion chapter. However, following discussion was added also to Dictra simulation chapter where decrease of Mn was modelled in lines 349-352:

"This in turn may explain partly the increase of the AF fraction in the CGHAZ of Tilowwith increasing cooling time as was presented in Figs. 5-7; driving force for the ferrite formation is known to increase along with decreasing Mn content. Similar observations have been achieved previously e.g. in [11–13,17]."

Best regards,

Henri Tervo

Reviewer 2 Report

The paper deals with the important topic of steels for offshore applications. The topic well fits with Metals journal purposes. 

The paper is clear, results clearly presented

I suggest the following improvement before publishing:

a)Line 45: add references.The following papers should be cited:

Lan, L., Chunlin, Q., Dewen, Z, XiuHau, G., Microstructural characteristich and toughness of the simulated coarse grained heat affected zone of high strength low carbon bainitic steels Materials Science and Engineering A 2011, 529, 192-200

Di Schino, A. Analysis of heat treatment effect on microstructural features evolution in a micro- alloyed martensitic steel. Acta Metall. Slovaca 2016, 22, 266–270.

b) Line 52 (after vicinity): please add reference

c) Line 62: authors adfirm "it has been reported": please add reference

d) Line 76:"Considerable amount of research into AF has been carried out in the past, especially for the weld...": please add references

e) Figure 5 (a-c-e):The markers are not readable

f) Figure 6 (a-c-e):The markers are not readable

g) Figure 7 (a-c-e):The markers are not readable

h) Figure 8 (a-c-e):The markers are not readable

i) Figure 9:The markers are not readable

l) Please check reference format: e.g. DOI code is not always present.

Author Response

Dear Editor,

The authors would like to thank the editor and the reviewers for their valuable comments and suggestions to improve the above manuscript. The response of the authors to the individual comments is given below and we hope that the changes are sufficient for the manuscript to be given final approval.

The amendments in the manuscript are marked using Track Changes function.

Reviewer’s comments:

Reviewer #2

Point 1: Line 45: add references.The following papers should be cited:

Lan, L., Chunlin, Q., Dewen, Z, XiuHau, G., Microstructural characteristich and toughness of the simulated coarse grained heat affected zone of high strength low carbon bainitic steels Materials Science and Engineering A 2011, 529, 192-200

Di Schino, A. Analysis of heat treatment effect on microstructural features evolution in a micro- alloyed martensitic steel. Acta Metall. Slovaca 2016, 22, 266–270.

Response 1: Added suggested references to line 45 (line 47 after other changes).

Point 2: Line 52 (after vicinity): please add reference

Response 2: Added some references to line 52 (line 54 after other changes).

Point 3: Line 62: authors adfirm "it has been reported": please add reference

Response 3: Added a reference to line 62 (line 64 after other changes).

Point 4: Line 76:"Considerable amount of research into AF has been carried out in the past, especially for the weld...": please add references

Response 4: Added references to line 76 (line 78 after other changes). 

Point 5: Figure 5 (a-c-e):The markers are not readable

Response 5: Figure 5 (a-c-e) modified to improve readability.

Point 6: Figure 6 (a-c-e):The markers are not readable

Response 6: Figure 6 (a-c-e) modified to improve readability.

Point 7: Figure 7 (a-c-e):The markers are not readable

Response 7: Figure 7 (a-c-e) modified to improve readability.

Point 8: Figure 8 (a-c-e):The markers are not readable

Response 8: Figure 8 (a-c-e) modified to improve readability.

Point 9: Figure 9:The markers are not readable

Response 9: Figure 9 modified to improve readability.

Point 10: Please check reference format: e.g. DOI code is not always present.

Response 10: References without DOI codes were checked manually and added DOI code when it was available. Some of the references are conference articles, standards, dissertations etc. and they don't have DOI code. Otherwise, reference list was made using Mendeley plug-in with the Metals journal style.

Best regards,

Henri Tervo

Reviewer 3 Report

REVIEW

on the article

Characterization of CGHAZ in Al- and Ti-deoxidized offshore steels

Henri Tervo, Antti Kaijalainen, Vahid Javaheri, Satish Kolli, Tuomas Alatarvas, Severi Anttila and Jukka Kömi

Summary.

The article is devoted to the urgent problem of studying various factors influencing the formation of acicular ferrite in coarse-grained heat-affected zone experimental laboratory steels. Existing steel grades operating in the harsh conditions of offshore oil fields do not fully meet the impact toughness specifications. High strength can lead to a decrease in toughness, especially after welding. The article rightly states that the structure should minimize the deterioration in toughness that occurs in the coarse-grained heat-affected zone. Thus, in this article, the authors investigate the characterization of coarse-grained heat-affected zone in three different laboratory steels, one deoxidized with Al79 and two deoxidized with Ti, which have chemical compositions that resemble steels used in the open sea.

This problem is urgent and of scientific interest.

The Introduction is presented by a review of references from 29 sources, including recent articles.

The results are well illustrated by photographs and figures of the obtained dependencies.

In Conclusion, the authors note that the optimal inclusion content for acicular ferrite was achieved with Tilow steel. In contrast, in Tihigh steel, the inclusions were generally too coarse, probably due to the high O and Ti contents.

In general, the article makes a good impression; the level of research is high and is of interest to researchers and readers.

However, the article has several flaws and unclearness.

Specific comments.

  1. I would recommend the authors remove the acronym CGHAZ from the title and include the coarse-grained heat-affected zone. It is more readable for readers.
  2. In Abstract authors tell that "Three experimental steels with comparable chemistry but differences in deoxidation practice were studied". Why? What is the principal scientific problem? Could you express in one sentence what the main problem is? And the reader's interest will rise!
  3. The Abstract does not reflect all the features of the article, so it must be redone. Editors strongly encourage authors to use the following style of structured abstracts, but without headings: (1) Background: Place the question addressed in a broad context and highlight the purpose of the study; (2) Methods: Describe briefly the main methods or treatments applied; (3) Results: Summarize the article's main findings; and (4) Conclusions: Indicate the main conclusions or interpretations. The abstract should be an objective representation of the article.

        I would recommend that the authors redo the Abstract and the Conclusion.

  1. The authors said "The specifications require that the degradation of material properties such as toughness in the weld heat-affected zone (HAZ) remain tolerable. Especially, the toughness degradation occurring in the coarse-grained heat-affected zone (CGHAZ), the intercritical heat-affected zone (ICHAZ), and the intercritically reheated coarse-grained heat-affected zone (ICCGHAZ) should be minimized by design".

       But the article does not answer this important question. How much will the toughness deteriorate? 1%, 5%, 10% or 50%? This specifies the importance of research. Compare the standard toughness to the toughness of the steels you studied, and it will be clear.

  1. I would recommend that the authors give (if possible) the mechanical characteristics of the steels under study: yield and strength limits, hardness, elongation, and impact strength. A quantitative comparison of the results obtained is most convincing.
  2. I recommend that the authors compare their results with those of other researchers in the field.

In general, the authors presented an interesting and relevant study which will undoubtedly be of interest to readers. However, there are many corrections, so I recommend the article for publication after major revision.

Author Response

Dear Editor,

The authors would like to thank the editor and the reviewers for their valuable comments and suggestions to improve the above manuscript. The response of the authors to the individual comments is given below and we hope that the changes are sufficient for the manuscript to be given final approval.

The amendments in the manuscript are marked using Track Changes function.

Reviewer’s comments:

Reviewer #3

Point 1: I would recommend the authors remove the acronym CGHAZ from the title and include the coarse-grained heat-affected zone. It is more readable for readers.

Response 1: The Acronym CGHAZ in the title changed to coarse-grained heat-affected zone as requested.

Point 2: In Abstract authors tell that "Three experimental steels with comparable chemistry but differences in deoxidation practice were studied". Why? What is the principal scientific problem? Could you express in one sentence what the main problem is? And the reader's interest will rise!

Response 2: Abstract was extended starting with the sentences "Deterioration of the toughness in heat-affected zones (HAZ) due to the thermal cycles caused by welding is a known problem in offshore steels. Acicular ferrite (AF) in the HAZ is generally considered beneficial regarding toughness." and continuing the cited sentence with "in order to find optimal conditions for the AF formation in the coarse-grained heat-affected zone (CGHAZ)."

Point 3: The Abstract does not reflect all the features of the article, so it must be redone. Editors strongly encourage authors to use the following style of structured abstracts, but without headings: (1) Background: Place the question addressed in a broad context and highlight the purpose of the study; (2) Methods: Describe briefly the main methods or treatments applied; (3) Results: Summarize the article's main findings; and (4) Conclusions: Indicate the main conclusions or interpretations. The abstract should be an objective representation of the article.

I would recommend that the authors redo the Abstract and the Conclusion.

Response 3: Abstract was formatted a bit as requested in Point 2 and consequently shortened to make it fit to the 200 word limit given in the manuscript guidelines. However, due to this limit it is very challenging to make the abstract to represent everything in the paper. Thus, only some main points were chosen to appear in the abstract.

Point 4: The authors said "The specifications require that the degradation of material properties such as toughness in the weld heat-affected zone (HAZ) remain tolerable. Especially, the toughness degradation occurring in the coarse-grained heat-affected zone (CGHAZ), the intercritical heat-affected zone (ICHAZ), and the intercritically reheated coarse-grained heat-affected zone (ICCGHAZ) should be minimized by design".

But the article does not answer this important question. How much will the toughness deteriorate? 1%, 5%, 10% or 50%? This specifies the importance of research. Compare the standard toughness to the toughness of the steels you studied, and it will be clear.

Response 4: The deteriorating of the toughness as well as beneficial effect of acicular ferrite on the toughness are based on literature solely. In this paper our aim is to focus on microstructural characterisation; in other words, trying to find optimal inclusion content, prior austenite grain size and cooling rate in this type of steel in order to produce acicular ferritic microstructure.

Point 5: I would recommend that the authors give (if possible) the mechanical characteristics of the steels under study: yield and strength limits, hardness, elongation, and impact strength. A quantitative comparison of the results obtained is most convincing.

Response 5: This paper is a part of the ongoing study. In this paper our aim is to focus on microstructural characterisation. The paper is rather wide already in current form, so adding more data would have easily made it unnecessarily large. This way we can also focus better on the mechanical properties in our next paper, when the microstructures have already been presented and analysed in this paper. Moreover, we don't even have all the mechanical tests performed yet, so also that is a reason to save that data for later use. However, hardness of different microstructures was already measured and presented in this paper to emphasise the differences of the obtained microstructures. Hardness values can also be used to approximate the ultimate tensile strength if necessary.

Point 6: I recommend that the authors compare their results with those of other researchers in the field.

Response 6: A bit more discussion together with references to some previous studies were added to chapter "DICTRA simulations" to lines 349-352 as follows:

"This in turn may explain partly the increase of the AF fraction in the CGHAZ of Tilowwith increasing cooling time as was presented in Figs. 5-7; driving force for the ferrite formation is known to increase along with decreasing Mn content. Similar observations have been achieved previously e.g. in [11–13,17]."

Best regards,

Henri Tervo

Round 2

Reviewer 1 Report

Unfortunately, this has not been fully modified yet. This manuscript is not suitable as a research paper because this only reports experimental results without deep discussion.

  1. Authors should discuss the reason why the fraction of AF increased with increasing cooling time (slow cooling rate) in Tilow sample. It is commonly accepted that the transformation of AF is situated between bainite and coarse grained ferrite. A higher cooling rate increases supercooling and therefore the reaction driving force, which leads to a higher nucleation rate.
  2. Authors should discuss the role of TiN containing inclusions. The microstructure of weld metals and heat affected zones is known to be refined by different inclusions, which may act as nucleation sites for intragranular acicular ferrite and/or to pin austenite grains thereby preventing grain growth. Please discuss the nature of inclusions and their influence on formation of AF.
  3. Please consider the change of the driving force quantitatively when the local decrease of Mn was observed in comparing to the change of interfacial energy.

Author Response

Dear Editor,

The authors would like to thank the editor and the reviewers for their valuable comments and suggestions to improve the above manuscript. The response of the authors to the individual comments is given below and we hope that the changes are sufficient for the manuscript to be given final approval.

The amendments in the manuscript are marked using Track Changes function.

Reviewer’s comments:

Reviewer #1

Point 1: Authors should discuss the reason why the fraction of AF increased with increasing cooling time (slow cooling rate) in Tilow sample. It is commonly accepted that the transformation of AF is situated between bainite and coarse grained ferrite. A higher cooling rate increases supercooling and therefore the reaction driving force, which leads to a higher nucleation rate.

Response 1: Following text was added to the lines 270-274:

The reason for the slight increase in the fraction of AF with increasing cooling time in Tilow stems with the continuous cooling transformation (CCT) diagram where the transformation of AF occurs between bainite and polygonal ferrite [33]. Thus, in the current study the increase in the cooling time reduced the driving force for the bainite transformation promoting AF transformation.

Point 2: Authors should discuss the role of TiN containing inclusions. The microstructure of weld metals and heat affected zones is known to be refined by different inclusions, which may act as nucleation sites for intragranular acicular ferrite and/or to pin austenite grains thereby preventing grain growth. Please discuss the nature of inclusions and their influence on formation of AF.

Response 2: The effect of TiN regarding the prior austenite grain growth was added to the lines 255-261 as follows:

”The enormous prior austenite grain growth in Tilow may suggest the lack of nanoscale precipitates such as TiN, which are known to inhibit the grain growth through the pinning effect. However, TiN was detected at least in microscale inclusions also in Tilow e.g. together with MnO-TiOand MnO-TiOx-MnS inclusions as was presented in Table 2. It is still evident that in Ref and Tihigh steels the fraction of inclusions containing considerable amount of TiN, such as pure TiN as well as MnS-TiN and MnO-TiN, was higher than in the Tilow steel. This may mean that also the number of nanoscale TiN was higher in Ref and Tihigh than in Tilow and that would explain the large PAGS in Tilow compared to other samples.”

However, the effect of TiN as nucleation sites for AF formation was neglected because according to the most references cited in the paper the most promising nucleation sites for AF formation are different type of titanium oxides instead of TiN. The role of inclusions on the formation of AF is already presented in the Introduction (lines 50-56) with relevant references as well as in the inclusion chapter of Results and Discussion (lines 212-224).

Point 3: Please consider the change of the driving force quantitatively when the local decrease of Mn was observed in comparing to the change of interfacial energy.

Response 3: I am not sure if I understood the comment appropriately. However, DICTRA considers the local gradient of Mn and thereby adjusts the driving force in solving the phase boundary problem. In other words, DICTRA considers the local equilibrium at the precipitate and matrix interface. This observed gradient in Mn is not related to the considered interfacial energy at the interface, rather depends on the growth kinetics of the MnS precipitate.

Best regards,

Henri Tervo

Reviewer 3 Report

REVIEW

on the article

Characterization of coarse-grained heat-affected zone in Al- and Ti-deoxidized offshore steels

Henri Tervo, Antti Kaijalainen, Vahid Javaheri, Satish Kolli, Tuomas Alatarvas, Severi Anttila and Jukka Kömi

Summary.

The article has been significantly revised, many ambiguities are well highlighted, the article looks much better. This article is clearly written and contains important new terms and explanations. The Abstract is revised, the background is marked, the main problem and the research results are highlighted.

The purpose of the study is clearly defined, the strengths of the article are shown quite clearly.

The authors did not respond to my comments on quantifying impact toughness and the effect of acicular ferrite on it. But I believe this will be the subject of further research by the authors.

In general, the article has been significantly improved and is of scientific interest.

I recommend the article for publication.

Author Response

Dear Editor,

The authors would like to thank the editor and the reviewers for their valuable comments and suggestions to improve the above manuscript. The response of the authors to the individual comments is given below and we hope that the changes are sufficient for the manuscript to be given final approval.

The amendments in the manuscript are marked using Track Changes function.

Reviewer’s comments:

Reviewer #3

Response: Thank you for the nice comments!

Best regards,

Henri Tervo

Round 3

Reviewer 1 Report

None.